# RECONCILING FEATURE-REUSE AND OVERFITTING IN DENSENET WITH SPECIALIZED DROPOUT

## ABSTRACT

Recently convolutional neural networks (CNNs) achieve great accuracy in visual recognition tasks. DenseNet becomes one of the most popular CNN models due to its effectiveness in feature-reuse. However, like other CNN models, DenseNets also face overfitting problem if not severer. Existing dropout method can be applied but not as effective due to the introduced nonlinear connections. In particular, the property of feature-reuse in DenseNet will be impeded, and the dropout effect will be weakened by the spatial correlation inside feature maps. To address these problems, we craft the design of a specialized dropout method from three aspects, dropout location, dropout granularity, and dropout probability. The insights attained here could potentially be applied as a general approach for boosting the accuracy of other CNN models with similar nonlinear connections. Experimental results show that DenseNets with our specialized dropout method yield better accuracy compared to vanilla DenseNet and state-of-the-art CNN models, and such accuracy boost increases with the model depth.

## 1 INTRODUCTION

Recent years have seen a rapid development of deep neural network in the computer vision area, especially for visual object recognition tasks. From AlexNet, the winner of ImageNet Large Scale Visual Recognition Challenge (ILSVRC) (Krizhevsky et al., 2012), to later VGG network (Simonyan & Zisserman, 2014) and GoogLeNet (Szegedy et al., 2015), CNNs have shown significant success with its shocking improvement of accuracy.

Researchers gradually realized that the depth of the network always plays an important role in the final accuracy of one model due to increased expressiveness (Håstad, 1987; Håstad & Goldmann, 1991; Szegedy et al., 2015). However, simply increasing the depth does not always help due to the induced vanishing gradient problem (Glorot & Bengio, 2010). ResNet (He et al., 2016a) has been proposed to promote the flow of information across layers without attenuation by introducing *identity skip-connections*, which sums together the input and the output of several convolutional layers. In 2016, Densely connected network (DenseNet) (Huang et al., 2016a) came out, which replaces the simple summation in ResNet with concatenation after realizing the summation may also impede the information flow.

Despite the improved information flow, DenseNet still suffers from the overfitting problem, especially when the network goes deeper. Standard dropout (Srivastava et al., 2014) has been used to combat such problem, but can not work effectively on DenseNet. The reasons are twofold: 1) Feature-reuse will be weakened by standard dropout as it could make features dropped at previous layers no longer be used at later layers. 2) Standard dropout method does not interact well with convolutional layers because of the spatial correlation inside feature maps (Tompson et al., 2015). Since *dense connectivity* increases the number of feature maps tremendously — especially at deep layers — the effectiveness of standard dropout would further be reduced.

In this paper, we design a specialized dropout method to resolve these problems. In particular, three aspects of dropout design are addressed: 1) Where to put dropout layers in the network? 2) What is the best dropout granularity? 3) How to assign appropriate dropout (or survival) probabilities for different layers? Meanwhile, we show that the idea to re-design the dropout method from the three aspects also applies to other CNN models like ResNet. The contributions of the paper can be summarized as follows:

- First, we propose a new structure named pre-dropout to solve the possible feature-reuse obstruction when applying standard dropout method on DenseNet.

- Second, we are the first to show that channel-wise dropout (compared to layer-wise and unit-wise) fit best for CNN through both detailed analysis and experimental results.

- Third, we propose three distinct probability schedules, and via experiments we find out the best one for DenseNet.

- Last, we provide a good insight for future practitioners, inspiring them regarding what should be considered and which is the best option when applying the dropout method on a CNN model.

Experiments in our paper suggest that DenseNets with our proposed specialized dropout method outperforms other comparable DenseNet and state-of-art CNN models in terms of accuracy, and following the same idea dropout methods designed for other CNN models could also achieve consistent improvements over the standard dropout method.

## 2 RELATED WORK

In this section, first we will review some basic ideas behind DenseNet, which is also the foundation of our method. Then the standard dropout method along with some of its variants will be introduced as a counterpart of our proposed dropout approach.

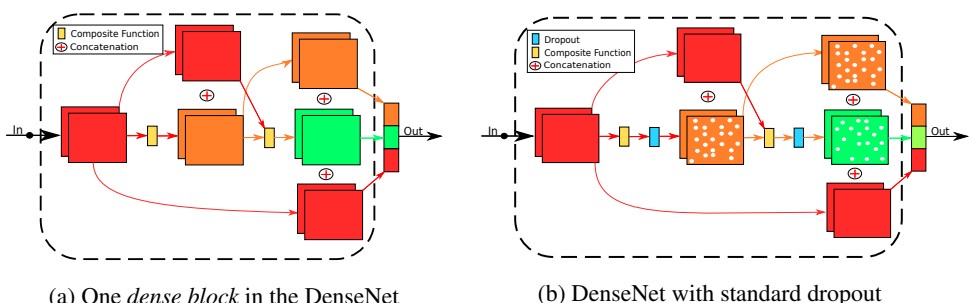

(a) One *dense block* in the DenseNet    (b) DenseNet with standard dropout

Figure 1: Examples of the *dense block* and the DenseNet with standard dropout method. White spots denote dropped neurons. The right figure shows that the same feature maps (orange color) after dropout layer will be directly sent to later layers, which makes the dropped features always unavailable to later layers.

### 2.1 DENSENET

DenseNet was first proposed by (Huang et al., 2016a), which features a special structure—*dense blocks*. Figure 1a gives an example of the *dense block*. One *dense block* consists of several convolutional layers, each of which output $k$ feature maps, where $k$ is referred to as the *growth rate* in the network. The most important property of DenseNet is that for each convolutional layer inside the *dense block*, the input is the concatenation of all feature maps from the preceding layers within the same block, which is also known as the *dense connectivity*. With *dense connectivity* previous output features could be reused at later layers.

$$X_\ell = L_\ell([X_0, X_1, \cdots, X_{\ell-1}]) \tag{1}$$

Equation 1 shows such kind of relationship clearly. Here $X_i$ represents the output at layer $i$. $[,]$ denotes concatenation operation. L is a composite function of batch normalization (BN) (Ioffe & Szegedy, 2015), rectified linear unit (ReLU) (Glorot et al., 2011) and a $3 \times 3$ convolutional layer.

With the help of feature-reuse, DenseNet achieves a better performance and turns out to be more robust (Huang et al., 2017). However, *dense connectivity* will cost a large amount of parameters. To ease the consumption of parameters, a variant structure, named DenseNet-BC, came out. In this structure, a $1 \times 1$ layer is added before each $3 \times 3$ layer to reduce the depth of input to $4k$.

### 2.2 DROPOUT

Standard dropout method (Hinton et al., 2012; Srivastava et al., 2014) discourages co-adaptation between units by randomly dropping out unit values. Stochastic depth (Huang et al., 2016b) extends

it to layer level by randomly dropping out whole layers. Other alternative examples include Drop-Connect (Wan et al., 2013), which generalizes dropout by dropping individual connections instead of units (i.e., dropping several connections together), and Swapout (Singh et al., 2016) which skips layers randomly at a unit level.

These previous works mainly focus on one aspect of dropout method, i.e., the dropout granularity. We are the first who gives a thorough study of the dropout design from all three aspects: dropout location, dropout granularity and dropout probability. In our evaluation, we will not only give the overall accuracy improvement, but also the breakdown along all these three aspects. Meanwhile, all these methods above will impede the feature-reuse in DenseNet since dropped features in previous layers will never be available to later layers. Figure 1b shows an example of the feature-reuse obstruction when applying standard dropout method on DenseNet.

# 3 MODEL DESCRIPTION

As previously mentioned, standard dropout method will have some limitations on DenseNet: 1. it could impede feature-reuse; 2. The effect of dropout will be weakened by the spatial correlation.

To solve these problems and further improve model generalization ability, we propose the following structures.

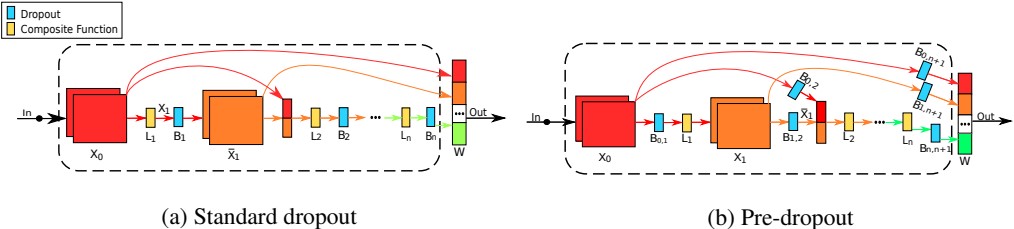

(a) Standard dropout        (b) Pre-dropout

Figure 2: Examples illustrating data flow in one *dense block* of standard dropout method and pre-dropout method. The blue boxes represent dropout layers, while the yellow boxes represent composite functions. $X_0$, $X_1$ and $\overline{X}_1$ are the data tensors. $W$ stands for the output of the *dense block*.

## 3.1 PRE-DROPOUT STRUCTURE

Pre-dropout structure aims at solving the possible feature-reuse obstruction when applying standard dropout to DenseNet. As mentioned in Section 2.2, due to the *dense connectivity*, standard dropout will make features dropped at previous layers no longer be used at later layers, which will weaken the effect of feature-reuse in DenseNet. To retain the feature-reuse, we come up with a simple yet novel idea named pre-dropout, which instead places dropout layers before the composite functions so that complete outputs from previous convolutional layers can be transferred directly to later layers before the dropout method is applied. Meanwhile one extra benefit from pre-dropout is that we can stimulate much more feature-reuse patterns in the network because of different dropout patterns applied before different layers. Figure 2 illustrates the differences between standard dropout and pre-dropout. In the following, we will explain these two benefits in details.

Suppose the input to Figure 2a is $X_0^{standard}$, then it would also be used as the input to the dropout layer $B_1$. The calculation of $B_1$ could be regarded as element-wisely multiplying $X_1^{standard}$ with $\Theta_1$, a tensor of random variables following $Bernoulli(p_1)$. Thus we can get

$$\overline{X}_1^{standard} = \Theta_1 \odot X_1^{standard} = \Theta_1 \odot L_1(X_0^{standard}) \tag{2}$$

where $\odot$ represents element-wise multiplication. Then $X_0^{standard}$ and $\overline{X}_1^{standard}$ will be concatenated together as the input to $L_2$, i.e., $X_0^{standard} \oplus \overline{X}_1^{standard}$, here $\oplus$ denotes concatenation. So on and so forth. Finally, the output of the *dense block* can be written as

$$W^{standard} = X_0^{standard} \oplus \overline{X}_1^{standard} \oplus \cdots \oplus \overline{X}_n^{standard} \tag{3}$$

Similarly we can get the mathematical representations from Figure 2b, which shows the data flow in a *dense block* with pre-dropout method,

$$\overline{X}_1^{pre} = \Theta_{1,2} \odot X_1^{pre} = \Theta_{1,2} \odot L_1(\Theta_{0,1} \odot X_0^{pre}) \tag{4}$$

$$W^{pre} = (\Theta_{0,n+1} \odot X_0^{pre}) \oplus (\Theta_{1,n+1} \odot X_1^{pre}) \oplus \cdots \oplus (\Theta_{n,n+1} \odot X_n^{pre}) \tag{5}$$

where $\Theta_{i,j}$ represents the tensor of dropout layer connecting from the output of layer $i$ to the input of layer $j$, in which random variables follow $Bernoulli(p_{i,j})$.

Comparing Equations 2 with 4, we can notice that pre-dropout method will allow better feature-reuse than standard dropout method. For instance, the outputs of standard dropout would become zero and remains the same as inputs to all following layers when $\Theta_1$ equals to zero. However pre-dropout solves this problem. In pre-dropout method, $X_1^{pre}$ would be multiplied by different independent tensors $\Theta_{1,2}, \Theta_{1,3}, \cdots, \Theta_{1,n+1}$, such that for any specific $X_{1ijk}^{pre}$ even if $\Theta_{1,2}$ makes $X_{1ijk}^{pre}$ be zero at $L_2$, we still have a chance to reuse this feature at later layers.

Meanwhile we could also find that pre-dropout method could stimulate much more feature-reuse patterns in the network. For example, in Figure 2a, once $X_1^{standard}$ goes through dropout layer $B_1$, the same output will always be reused. Whereas in pre-dropout method, every time before $X_1^{pre}$ is utilized as the input to the next layer, it would be multiplied by a different tensor. In Figure 2b, the contributions of $X_1^{pre}$ are actually two distinct features, since $\Theta_{1,2}$ and $\Theta_{1,n+1}$ are independent.

Note that similar feature-reuse obstruction also exists when applying standard dropout on other CNN models with shortcut connections, such as ResNet (He et al., 2016a), Wide-ResNet (Zagoruyko & Komodakis, 2016) and RoR (Zhang et al., 2017). Thus pre-dropout method could work for those networks as well.

## 3.2 CHANNEL-WISE DROPOUT

When designing our specialized dropout method, dropout granularity is also an important aspect to be considered. Figure 3 shows three different granularity: unit-wise, channel-wise and layer-wise.

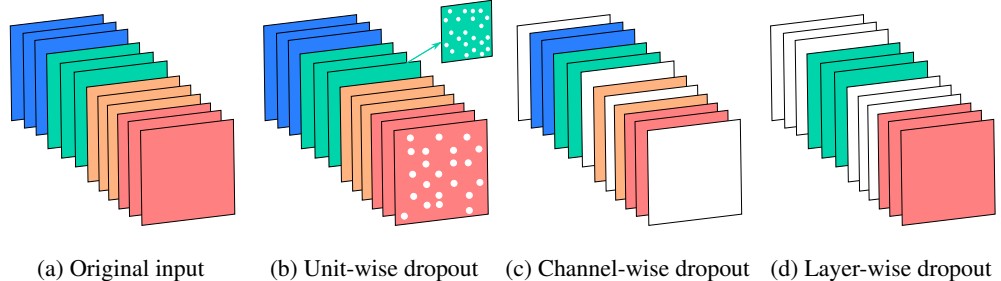

|                   |                      |                         |                      |
|:-----------------:|:--------------------:|:-----------------------:|:--------------------:|
| (a) Original input | (b) Unit-wise dropout | (c) Channel-wise dropout | (d) Layer-wise dropout |

Figure 3: Different dropout granularity, white color represents dropout.

Standard dropout is one kind of unit-wise method, which has been proved useful when applied on fully connected layers (Srivastava et al., 2014). It helps improve the generalization capability by breaking the strong dependence between neurons from different layers.However when applying the same method on convolutional layers, the effectiveness of standard unit-wise method will be hampered due to the strong spatial correlation between neurons within a feature map — although dropping one neuron can stop other neurons from replying on that particular one, they will still be able to learn from the correlated neurons in the same feature map.

To cope with the spatial correlation, we need dropout at better granularity. Layer-wise method drops the entire layers, whcih refer to the outputs from previous layers, inside the input tensor. However, layer-wise dropout is prone to discard too many useful features. Channel-wise method, which will drop a entire feature map at a given probability, strikes a nice trade-off between the two granularity above. Our experiments also confirmed that channel-wise works the best for regularizing DenseNet.

Meanwhile since the spatial correlation exists in all types of convolutional layers, our analysis above should also work for other CNN models whenever a dropout method is applied.

## 3.3 PROBABILITY SCHEDULE

Channel-wise dropout can still be improved when applied on DenseNet. Notice that naive channel-wise dropout cannot add various degrees of noise to different layers due to the deterministic survival

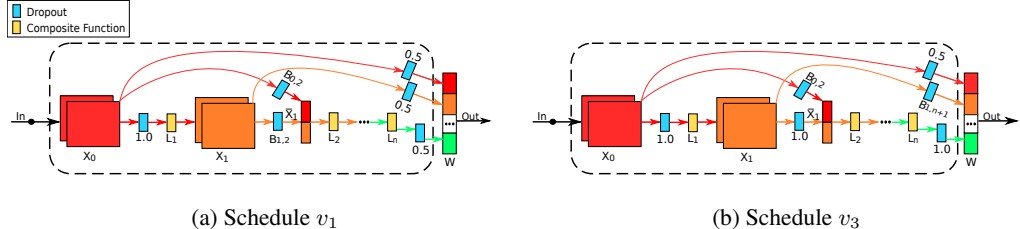

(a) Schedule $v_1$          (b) Schedule $v_3$

Figure 4: Different probability schedules. The numbers besides blue boxes represent survival probabilities. Left figure shows the linearly decaying schedule $v_1$ which applies linearly decaying probabilities on different layers of DenseNet. Right figure shows $v_3$ schedule which applies various probabilities on different portions of the input to a convolutional layer depending on distances between layers generating those portions and the input layer.

probability, and since in DenseNet *dense connectivity* makes the sizes of inputs at different layers quite different, such variation seems to be helpful. We believe the model would benefit from such kind of noise. Thus, in our design, to further promote model generalization ability, besides predropout and channel-wise dropout, we also apply stochastic probability method to introduce various degrees of noise to different layers in DenseNet. In experiment part we compare the stochastic probability method with the deterministic probability method, and results show that stochastic method could get a better accuracy on DenseNet. Since in CNNs the degree of feature correlation and the importance of features are generally different at different layer depths, such probability schedule would always be desired.

Once we adopt the stochastic method, one natural question arises: *how can we assign probabilities for different convolutional layers in order to achieve the best accuracy?* Actually it is hard to find out the best specific probability for each layer. However, based on some observations we are still able to design some useful probability schedules. Figure 4 gives examples on the different schedules below. Same as before, we've done some experiments on them and adopt the best one in our design.

One observation is that in the shallow layers of DenseNet, the number of feature maps used as input is quite limited and as the layers go deeper the number will become larger. Meanwhile in CNN highlevel features are prone to be repeated. Thus intuitively we propose a linearly decaying probability schedule. We refer to it as $v_1$. For this probability schedule, in each *dense block*, the starting survival probability at the first convolutional layer is 1.0 while the last one is 0.5. Recall that in Section 3.1 we use $\Theta_{i,j}$ to represent the tensor of dropout layer connecting from the output of layer $i$ to the input of layer $j$, in which random variables follow $Bernoulli(p_{i,j})$. So schedule $v_1$ will have the following properties,

1. For fixed $j$ and $\forall i$, $p_{i,j} = $ C, C is a constant. Particularly, $p_{i,1} = 1.0$ and $p_{i,n+1} = 0.5$;
2. For fixed $i$, $p_{i,j}$ is monotonically decreasing with $j$.

To the best of our knowledge, dropout will add per neuron different levels of stochasticity depending on the survival probability and maximum randomness is reached when probability is 0.5. Meanwhile the sizes of inputs in DenseNet will gradually increase. So to reduce the total randomness in the model, we design another schedule $v_2$, which is also a reverse version of $v_1$, i.e., the starting probability for the first layer is 0.5 whereas the last one is 1.0. Similarly, in this schedule, we will have,

1. For fixed $j$ and $\forall i$, $p_{i,j} = $ C, C is a constant. Particularly, $p_{i,1} = 0.5$ and $p_{i,n+1} = 1.0$;
2. For fixed $i$, $p_{i,j}$ is monotonically increasing with $j$.

Additionally, we observe that in DenseNet deeper layers tend to rely on high-level features more than low-level features, such phenomenon is also mentioned in (Huang et al., 2016a). Based on that, schedule $v_3$ is proposed. For this schedule, we decide survival probabilities for different layers' outputs based on their distances to the convolutional layer, i.e., the most recent output from previous layers will be assigned with the highest probability while the earliest one gets the lowest. In our implementation, for the input to the last layer in one *dense block* we assign the most recent output in it with probability 1.0 and the least with 0.5. Then based on the number of previous layers concatenated, we can calculate the probability difference between two adjacent layers' outputs to decide probabilities for other portions of the input. The corresponding properties of $v_3$ can be summarized as,

1. $p_{i,j} \propto \frac{1}{d(i,j)}$, where $d(i,j)$ denotes the distance between layer $i$ and $j$;

2. For fixed $j$, $p_{i,j}$ is monotonically increasing with $i$. Particularly, when $i = j-1$, $p_{i,j} = 1.0$, and $p_{0,n+1} = 0.5$;

3. For fixed $i$, $p_{i,j}$ is monotonically decreasing with $j$. Particularly, when $j = i+1$, $p_{i,j} = 1.0$.

In conclusion, by using $v_3$ for inputs to deep layers in DenseNet low-level features from shallow layers will always be dropped with higher probabilities whereas high-level features can be kept with a good chance. Meanwhile, survival probabilities for the output of one layer will become smaller as layers go deeper, which is also intuitive as outputs from earlier layers have been used for more times so there exists higher probability that such outputs will not be used again later.

Also notice that the idea to apply different dropout probability at different layers can also be applied to other networks since the variation of inputs at different layers are quite common CNN models.

Table 1: Test errors (%) of various network structures and methods

| Structure and method | Depth | Params | C10 | C100 |
|---|---|---|---|---|
| FitNet (Romero et al., 2014) | 19 | - | 8.39 | 35.04 |
| Deeply Supervised Net (Lee et al., 2015) | - | - | 7.97 | 34.57 |
| Highway Network (Srivastava et al., 2015) | - | - | 7.72 | 32.39 |
| ResNet v1 with Stochastic Depth (Huang et al., 2016b) | 110 | 1.7M | 5.23 | 24.58 |
| ResNet v2 (He et al., 2016b) | 164 | 1.7M | 5.46 | 24.33 |
| ResNet v2 (He et al., 2016b) | 1001 | 10.2M | 4.92 | 22.71 |
| Swapout v2 $W \times 2$ (Singh et al., 2016) | 20 | 1.09M | 5.68 | 25.86 |
| Swapout v2 $W \times 4$ (Singh et al., 2016) | 32 | 7.46M | 4.76 | 22.72 |
| DenseNet-BC | 76 | 0.5M | 5.21 | 24.09 |
| DenseNet-BC(standard dropout) | 76 | 0.5M | 5.56 | 24.75 |
| DenseNet-BC(our specialized dropout) | 76 | 0.5M | 4.94 | 23.90 |
| DenseNet-BC | 100 | 0.8M | 4.73 | 23.22 |
| DenseNet-BC(standard dropout) | 100 | 0.8M | 5.01 | 23.80 |
| DenseNet-BC(our specialized dropout) | 100 | 0.8M | 4.51 | 22.33 |
| DenseNet-BC | 148 | 1.5M | 4.31 | 20.76 |
| DenseNet-BC(standard dropout) | 148 | 1.5M | 4.60 | 22.28 |
| DenseNet-BC(our specialized dropout) | 148 | 1.5M | **3.90** | **19.75** |

## 4 EXPERIMENTS

### 4.1 DATASETS AND EXPERIMENTAL SETTINGS

In our experiments, we mainly use two datasets: CIFAR10 and CIFAR100, containing 50k training images and 10k test images, a perfect size for a model of normal size to overfit. Meanwhile, we apply normal data augmentation on them which includes horizontal flipping and translation.

When implementing our models, we try to retain the same configurations of original DenseNet though further hyper-parameter tuning might generate better results since dropout will slow convergence (Srivastava et al., 2014). Briefly four DenseNet structures are used: DenseNet-40, DenseNet-BC-76, DenseNet-BC-100 and DenseNet-BC-147. The number at the rear of the structure name represents the depth of the network. Growth rate $k$ for all structures is 12. We adopt 300 training epochs with batch size 64. The initial learning rate is 0.1 and is divided by 10 at 50% and 75% of the total number of training epochs. During training process a fixed survival probability 0.5 is used for non-stochastic dropout methods. The test error is reported after every epoch and all results in our experiments represent test errors after the final epoch.

### 4.2 OVERALL EFFECTIVENESS OF SPECIALIZED DROPOUT

As an important part of our work, we want to know how DenseNets with our specialized dropout method compare with other models. In this section, we use three different DenseNet structures and test on CIFAR10 and CIFAR100 augmentation datasets.

Note that our specialized dropout won't incur additional parameters in the network. From results in Table 1, we can find that DenseNets with our specialized dropout method get the best accuracy on both datasets. In particular specialized dropout could consistently have good improvements over standard dropout method and DenseNet-BC-100 with our specialized dropout which only contains 0.8M parameters could outperform a 1001 layer ResNet model. Notice that data augmentation (which is always applied) already imposes generalization power to the model, which could make other regularization methods less effective.

Further, based on the results, our specialized dropout method gives better accuracy improvements on larger DenseNets, e.g., on CIFAR100 dataset, the improvements of our specialized dropout method would increase with the depth of the network.

### 4.3 EXPERIMENTS ON PRE-DROPOUT STRUCTURE

In order to show the effectiveness of pre-dropout structure, we compare unit-wise pre-dropout method with standard dropout on two DenseNet structures. We run experiments on CIFAR10 augmentation dataset. Results are shown in Table 2.

From Table 2, pre-dropout structure achieves better accuracy than standard dropout on both of the two DenseNet structures. Such results coincide with our analysis that pre-dropout could incur better feature-reuse and stimulate various features in the network. Meanwhile according to the analysis in Section 3.2 one factor that could disadvantage pre-dropout here is that correlated features can compensate for dropped features in standard dropout method.

Table 2: Unit-wise pre-dropout vs standard dropout

| | Test error (%) | |
|---|---|---|
| Structure | standard dropout | Pre-dropout |
| DenseNet-40 | 5.83 | **5.75** |
| DenseNet-BC-76 | 5.56 | **5.25** |

### 4.4 EFFECTIVENESS OF CHANNEL-WISE DROPOUT

In this section, we use DenseNet-BC-76 structure to compare the three dropout granularity on CIFAR10 and CIFAR100 augmentation datasets respectively. The results are shown in Table 3.

Table 3 shows that channel-wise dropout achieves the best accuracy on both of two datasets. So in our specialized dropout method, we adopt channel-wise dropout granularity. Also from Table 3, layer-wise dropout always has the worst performance. The reason could be that layer-wise dropout discards some useful features at one time, as a result a loss of accuracy is observed.

Table 3: Comparisons of different dropout granularity

| | Test error (%) | |
|---|---|---|
| Method | C10 | C100 |
| Unit-wise dropout | 5.25 | 24.52 |
| Layer-wise dropout | 5.46 | 25.28 |
| Channel-wise dropout | **5.09** | **24.36** |

### 4.5 EXPERIMENTS ON DIFFERENT PROBABILITY SCHEDULES

In Section 3.3, we argue why DenseNet would benefit from the variation of noise at different layers. In order to validate this idea, we compare the proposed three stochastic probability schedules to the standard version with a uniform dropout probability (0.5) on DenseNet-BC-76. Our empirical study indicates that $v_3$ always is the best among the three schedules, whereas $v_2$ is the worst. Thus, we pick schedule $v_3$ as our final specialized dropout method.

Table 4: Comparisons of various probability schedules

| Method | Error (%) |
|---|---|
| Channel-wise dropout (uniform 0.5) | 5.09 |
| Channel-wise dropout with $v_1$ | 5.02 |
| Channel-wise dropout with $v_2$ | 5.13 |
| Channel-wise dropout with $v_3$ | **4.94** |

Table 4 gives an example result for DenseNet-BC-76 on CIFAR10 augmentation dataset. Recall that the number of feature maps to shallow layers in DenseNet is very limited and schedule $v_2$ applies lower survival probabilities on these layers, thus from the results we can see although $v_2$ reduces the total randomness in the model, a loss of relatively larger quantity of low-level features could still hurt the accuracy.

Furthermore, we can find that the same effect of $v_1$ also exists in $v_3$, i.e., relatively higher survival probabilities are assigned for shallow layers and lower ones for deep layers. Besides $v_3$ can also help deep layers rely more on high-level features, which could be the reason making $v_3$ better than $v_1$.

## 4.6 REGULARIZATION EFFECT

We also want to figure out the reasons why our specialized dropout method could result in an accuracy improvement. To reveal the reasons, we compare the training/test errors during the training procedures of the normal DenseNet and the one with our specialized dropout method. Figure 5 shows such comparison on DenseNet-BC-148. As shown in the figure, the specialized dropout version reaches slightly higher training error at convergence, but produces lower test error. This phenomenon indicates that the improvement of accuracy comes from the strong regularization effect brought by the specialized dropout method, which also verifies that the specialized dropout method could improve the model generalization ability.

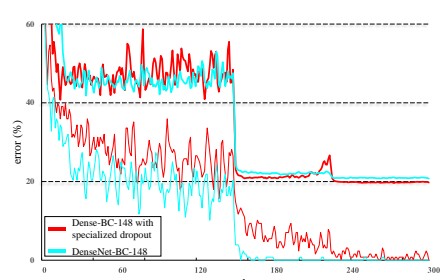

Figure 5: Training on CIFAR100. Thin curves denote training error, and bold curves denote test error.

## 4.7 SPECIALIZED DROPOUT ON OTHER CNN MODELS

Following the idea of designing a specialized dropout method for DenseNet, we also want to explore whether such idea could also apply to other state-of-the-art CNN models. Here we choose AlexNet, VGG-16 and ResNet to conduct the experiments. Similar to the DenseNet, we design the specialized dropout method for each model from three aspects. We apply pre-dropout structure and channel-wise granularity for all specialized dropout methods and decide dropout probability by the size of input. In order to reduce the total randomness in a model, the largest input will have the dropout probability 0 while the smallest one corresponds to the probability 0.5. Layers between the two will follow a linear increasing/decreasing schedule to assign the dropout probability. The results are shown in Table 5.

Table 5: Effectiveness of the specialized dropout method on other state-of-the-art CNN models. The model names in bold denote models with the specialized dropout method. Results are reported as test errors.

| Model | Depth | C10 | C100 |
|---|---|---|---|
| AlexNet | 8 | 10.32 | 38.71 |
| **AlexNet** | 8 | 9.78 | 35.42 |
| VGG-16 | 16 | 8.39 | 35.04 |
| **VGG-16** | 16 | 7.25 | 33.71 |
| ResNet v1 | 110 | 6.41 | 27.22 |
| **ResNet v1** | 110 | 5.45 | 25.46 |
| ResNet v2 | 164 | 5.46 | 24.33 |
| **ResNet v2** | 164 | 4.38 | 23.36 |

From results in Table 5 we can see that models with the specialized dropout method all outperform its original counterparts, which indicates that our idea to design a specialized dropout method could also work in other CNN models. The effectiveness of our idea is also validated by such results.

## 5 CONCLUSION

In this paper, first we show problems of applying standard dropout method on DenseNet. To deal with these problems, we come up with a new pre-dropout structure and adopt channel-wise dropout granularity. Specifically, we put dropout before convolutional layers to reinforce feature-reuse inside the model. Meanwhile we randomly drop some feature maps in inputs of convolutional layers to break dependence among them. Besides to further promote model generalization ability we introduce stochastic probability method to add various degrees of noise to different layers in DenseNet. Experiments show that in terms of accuracy DenseNets with our specialized dropout method outperform other CNN models.

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
