# OpenReview forum: "Reconciling Feature-Reuse and Overfitting in DenseNet with Specialized Dropout"
_ICLR.cc/2019/Conference_

### Official Review · AnonReviewer2 · 2018-11-01
**Evaluation of dropout regimes for DenseNets**

**Rating:** 4
**Confidence:** 4

**Review:**

The paper studies the effect of different dropout regimes (unit-wise, channel-wise and layer-wise), locations and probability affect the performance of DenseNet classification model. The experiments are performed on two datasets: CIFAR10 and CIFAR100.

In order to improve the paper, the authors could take into consideration the following points:

1. The experimental validation is rather limited. Additional experiments on large scale datasets should be performed (e. g. on ImageNet).
2. The design choices are rather arbitrary. The authors study three different probability schedules. Wouldn't it be better to learn them using recent advances in neural architecture search or in RL.
3. "The test error is reported after every epoch and ...". This suggest that the authors are monitoring the test set throughout the training. Thus, the hyper parameters selected (e. g. the dropout regimes) might reflect overfitting to the test set.
4. Table 1 misses some important results on CIFAR10 and CIFAR100, as is, the Table suggest that the method described in the paper is the best performing method on these datasets (and it is not the case). Moreover, the inclusion criteria for papers to appear in Table 1 is not clear. Could the authors correct the Table and add recent results on CIFAR10 and CIFAR100?
5. Section 4.1: "... a perfect size for a model of normal size to overfit."  This statement is not clear to me. What is a normal size model? Moreover, claiming that CIFAR10 and CIFAR100 is of perfect size to overfit seems to be a bit misleading too. Please rephrase.
6. Section 3.3: what do the authors mean by deterministic probability model?
7. Abstract: "DenseNets also face overfitting problem if not severer". I'm not aware of any evidence for this. Could the authors add citations accordingly?
8. Some discussions on recent approaches to model regularizations and connections to proposed approach are missing. The authors might consider including the following papers: https://arxiv.org/pdf/1708.04552.pdf, https://arxiv.org/pdf/1802.02375.pdf, among others.

Overall, the paper is easy to understand. However, the originality of the paper is rather limited and it is not clear what is the added value to for the community from such paper. I'd encourage the authors to include additional experiments, correct misleading statements and add a discussion of model regularization techniques in the related work section.

---

### Official Review · AnonReviewer1 · 2018-11-04
**interesting heuristics but no justification either theoretically or empirically**

**Rating:** 3
**Confidence:** 3

**Review:**

This paper proposes a special dropout procedure for densenet. The main argument is standard dropout strategy may impede the feature-reuse in Densenet, so the authors propose a pre-dropout technique, which implements the dropout before the nonlinear activation function so that it can be feeded to later layers. Also other tricks are discussed, for example, channel-wise dropout, and probability schedule that assigns different probabilities for different layers in a heuristic way.

To me this is a mediocre paper. No theoretical justification is given on why their pre-dropout structure could benefit compared to the standard dropout. Why impeding the feature-reuse in the standard dropout strategy is bad? Actually I am not quite sure if reusing the features is the true reason densenet works well in applications.

Heuristic is good if enough empirical evidence is shown, but I do not think the experiment part is solid either. The authors only report results on CIFAR-10 and CIFAR-100. Those are relatively small data sets. I would expect more results on larger sets such as image net.

Cifar-10 is small, and most of the networks work fairly well on it. Showing a slight improvement on CIFAR-10 (less than 1 point) does not impress me at all, especially given the way more complicated way of the dropout procedure.

The result of the pre-dropout on CIFAR-100 is actually worse than the original densenet paper using standard dropout. Densenet-BC (k=24) has an error rate of 19.64, while the pre-dropout is 19.75.

Also, the result is NOT the-state-of-the-art. Wide-ResNet with standard dropout has better result on both CIFAR-10 and CIFAR-100, but the authors did not mention it.

---

### Official Review · AnonReviewer3 · 2018-11-06
**This paper concerns the application of different binary dropout structures and schedules with the specific aim to regularise the DenseNet architecture.**

**Rating:** 5
**Confidence:** 4

**Review:**

Overall Thoughts:

I think the use of regularisation to improve performance in DenseNet architectures is a topic of interest to the community. My concern with the paper in it’s current form is that the different dropout structures/schedules are priors and it is not clear from the current analysis exactly what prior is being specified and how to match that to a particular dataset. Further, I believe that the current presentation of the empirical results does not support the nature of the claims being made by the authors. I would be very interested to hear the authors’ comments on the following questions.

Specific Comments/Questions:

Sec1: Sorry if I have missed something but for the two reasons against std dropout on dense net, the reference supports the second claim but could a reference be provided to substantiate the first?

Sec1/2: The discussion around feature re-use needs to be clarified slightly in my opinion. Dropout can provide regularisation in a number of regimes - the term “feature reuse” is a little tricky because I can see the argument from both sides - under the authors arguments, forcing different features to be used can be a source or robustness so would not the level of granularity be something to be put in as a prior and not necessarily inherently correct or incorrect?

Sec3: The key contribution (in my opinion) suggested by the authors is the “detailed analysis” of their dropout structures. I’m afraid I didn’t see this in this section - there are a number of approaches that have been taken in the literature to analyse the regularisation properties of dropout - e.g. the insightful approach of Gal and Ghahramani on dropout as a Bayesian regulariser (as well as others). I was expecting to see something similar to this - could the authors comment on this? Would such an analysis be possible - it would reveal the true priors being applied by the different approaches and allow an analysis of the priors being applied by the different methods?

Sec3: Similarly, with the dropout probability schedules, there are practical methods for learning such probabilities during training (e.g. Concrete Dropout) - would it not be possible to learn these parameters with these approaches? Why do we need to set them according to fixed schedules? I think it would be necessary to demonstrate that a fixed schedule outperforms learned parameters.

Sec4: My main difficulty here is that the other key contribution of the paper are the claims constructed around empirical results. Throughout the results section, only single values are presented without attempt to measure the distributions of the results (not even error bars). Without this information it is impossible to make any statements on the significance of the results. Ideally histograms should be provided (rather than just error bars). How do we know the changes conferred are significant for the particular problems? How do we know that they are causal from the new structures and not from hyper parameters or optimisation effects?

Sec4: Dropout is the application of a prior - how do we know what this prior is doing and when it is sensible to apply it? How do we know the results will transfer to datasets other than CIFAR?

Sec4: Please could the authors provide justification to the claim that the improvements would increase with the depth of the network?

Refs: Please could the authors be sure to cite the published versions of articles (not ArXiv versions) when papers have been peer reviewed - e.g. the citation for DenseNet (among others)

Other Points:

Could the authors use text mode for sub or superscripts in maths equations when using words as opposed to symbols?

There are a number of uses of “could” when I don’t think the authors mean “could” - please could this be checked?

Typos:

p4 replying -> relying, whcih -> which

---

### Meta-Review · Area_Chair1 · 2018-12-10

**Confidence:** 5
**Recommendation:** Reject

**Metareview:**

All reviewers recommend reject and there is no rebuttal. There is no basis on which to accept the paper.